# Sample-Then-Optimize Batch Neural Thompson Sampling

**Zhongxiang Dai[†], Yao Shu[†], Bryan Kian Hsiang Low[†], Patrick Jaillet[§]**
Dept. of Computer Science, National University of Singapore, Republic of Singapore[†]
Dept. of Electrical Engineering and Computer Science, MIT, USA[§]
{daizhongxiang,shuyao,lowkh}@comp.nus.edu.sg[†],jaillet@mit.edu[§]

## Abstract

*Bayesian optimization* (BO), which uses a *Gaussian process* (GP) as a surrogate to model its objective function, is popular for black-box optimization. However, due to the limitations of GPs, BO underperforms in some problems such as those with categorical, high-dimensional or image inputs. To this end, recent works have used the highly expressive *neural networks* (NNs) as the surrogate model and derived theoretical guarantees using the theory of *neural tangent kernel* (NTK). However, these works suffer from the limitations of the requirement to invert an extremely large parameter matrix and the restriction to the sequential (rather than batch) setting. To overcome these limitations, we introduce two algorithms based on the *Thompson sampling* (TS) policy named *Sample-Then-Optimize Batch Neural TS* (STO-BNTS) and STO-BNTS-Linear. To choose an input query, we only need to train an NN (resp. a linear model) and then choose the query by maximizing the trained NN (resp. linear model), which is equivalently sampled from the GP posterior with the NTK as the kernel function. As a result, our algorithms sidestep the need to invert the large parameter matrix yet still preserve the validity of the TS policy. Next, we derive regret upper bounds for our algorithms with batch evaluations, and use insights from batch BO and NTK to show that they are *asymptotically no-regret* under certain conditions. Finally, we verify their empirical effectiveness using practical AutoML and reinforcement learning experiments.

## 1 Introduction

*Bayesian optimization* (BO), also called *Gaussian process* (GP) bandits, has become a celebrated method for optimizing expensive-to-compute black-box functions, primarily thanks to its practical sample efficiency and theoretically guaranteed convergence [11, 14, 21, 51]. However, there are important problem settings where BO either underperforms or is not even applicable without sophisticated modifications, such as problems with categorical [18], high-dimensional [30], or images inputs. These issues have arisen mainly because GPs (i.e., the surrogate used by BO to model the objective function) are not able to effectively model these types of input space, which therefore calls for the use of alternative surrogate models in BO. To this end, *neural networks* (NNs) serve as a natural candidate owing to their remarkable expressivity [37]. NNs have repeatedly proven their ability to model extremely complicated real-world functions such as those involving categorical, high-dimensional or image inputs, whereas the development of GPs to effectively model these functions still represents active areas of research. In this regard, [69] have adopted NNs as the surrogate model in contextual bandit problems and employed the theoretical framework of *neural tangent kernel* (NTK) [27] to construct a principled algorithm following the well-known policy of *upper confidence bound* (UCB), hence introducing the *Neural UCB* algorithm. More recently, [66] have extended Neural UCB to

---

Correspondence to: Yao Shu <shuyao@comp.nus.edu.sg>.

36th Conference on Neural Information Processing Systems (NeurIPS 2022).

follow the *Thompson sampling* (TS) policy and proposed the *Neural TS* algorithm. Both Neural UCB and Neural TS are equipped with upper bounds on their cumulative regret and perform competitively in real-world contextual bandit experiments.

However, Neural UCB and Neural TS are still faced with several important limitations that may hinder their practical applications. Firstly, these algorithms suffer from the requirement to invert a $p \times p$ matrix in every iteration, in which $p$ is the number of parameters of the NN surrogate and is usually extremely large since the theory of NTK requires severe overparameterization. In practice, a diagonal approximation is used to avoid the need to invert such a large $p \times p$ matrix [66, 69], which introduces approximation errors in their practical deployment that are unaccounted for in the theoretical analysis, hence causing a disparity between theory and practice. Secondly, the theoretical analyses of these algorithms are restricted to the sequential setting and are hence not applicable in the batch setting where an entire batch of inputs are selected for querying.[1] To overcome these two limitations, we introduce two algorithms based on the TS policy named *Sample-Then-Optimize Batch Neural Thompson Sampling* (STO-BNTS) and STO-BNTS-Linear, both of which *(1)* sidestep the need to invert the $p \times p$ matrix and hence close the gap between theory and practice, and *(2)* naturally support batch evaluations while preserving the theoretical guarantees.

To avoid the inversion of the $p \times p$ matrix while still ensuring the validity of the TS policy, we draw inspirations from sample-then-optimize optimization [42] and Bayesian deep ensembles [24] to efficiently sample functions from the GP posterior (with the NTK as the kernel function) without inverting the $p \times p$ matrix. Specifically, to choose an input query within a batch, our STO-BNTS (resp. STO-BNTS-Linear) only needs to use the current observation history to train an *NN surrogate* (resp. a linear model defined w.r.t. the random features embedding of the NTK associated with an *NN surrogate*) using randomly initialized parameters, and then choose the input that maximizes the trained NN (resp. trained linear model). As a result, if the NN surrogate has an infinite width, the function of the trained NN (resp. trained linear model) is equivalently sampled from the GP posterior with the NTK as the kernel function. This ensures that our algorithms follow the TS policy and hence lays the foundation for our theoretical analyses. Next, to address the second challenge of deriving theoretical guarantees for our algorithms in the batch setting, we generalize the theoretical analysis of sequential TS [7] to account for batch evaluations and derive a regret upper bound for both of our algorithms when the NN surrogate is infinite-width. Then, we leverage insights from batch BO [17] and NTK [31] to show that the regret upper bound is sub-linear (under some conditions), which implies that our algorithms are *asymptotically no-regret*. Next, when the NN surrogate is *finite-width*, we derive a regret upper bound for our STO-BNTS-Linear by carefully accounting for the approximation error caused by the use of a finite (instead of infinite) NN, and show that the regret upper bound of STO-BNTS-Linear remains sub-linear as long as the NN is wide enough.

Our contributions are summarized as follows:

- Our STO-BNTS and STO-BNTS-Linear algorithms sidestep the inversion of the $p \times p$ matrix required by Neural UCB and Neural TS, which closes their gap between theory and practice.
- Our algorithms naturally support batch evaluations with theoretical guarantees.
- Our algorithms are equipped with an upper bound on their cumulative regret when the NN surrogate is infinite-width, and are asymptotically no-regret (i.e., their regret upper bound is sub-linear) under certain conditions. Moreover, when the NN surrogate is finite-width, our STO-BNTS-Linear still enjoys a regret upper bound, which remains sub-linear as long as the NN is wide enough.
- We demonstrate our empirical effectiveness in real-world experiments including *automated machine learning* (AutoML) and *reinforcement learning* (RL) tasks, as well as a task on optimization over images. To the best of our knowledge, our experiments (Sec. 5) are the first empirical study to show the advantage of neural bandit over GP bandit algorithms in practical AutoML and RL tasks.

## 2    Background

**Neural Networks and Neural Tangent Kernel.** In this work, we adopt the same construction of a neural network (NN) as [2]. We use $f(\mathbf{x}; \theta)$ to denote the scalar output of an $(L+1)$-layer NN with parameters $\theta \in \mathbb{R}^p$ and input $\mathbf{x}$, and use $\nabla_\theta f(\mathbf{x}; \theta')$ to represent the gradient of the NN evaluated at

---

[1]The work of [23] focuses on the *contextual bandit* setting and aims to choose a batch of inputs *given a batch of diverse contexts*. So, their methods are not applicable in our setting of BO where the contexts are fixed in all iterations, since they do not explicitly encourage input diversity which is a crucial problem in batch BO [8, 17].

$\theta = \theta'$. For simplicity, we assume every layer of the NN has the same width and represent the width as $m$. We denote our initializaiton scheme for $\theta$ as $\mathrm{init}(\cdot)$ which simply independently samples every NN parameter from the standard Gaussian distribution. The NTK [27] provides an explicit connection between NNs trained via gradient descent and kernel regression using NTK as the kernel function [2]. The NTK matrix, denoted as $\Theta$, has been shown to stay constant during the course of training as the width $m$ of the NN approaches infinity [27]. Moreover, $\Theta$ can be approximated by an *empirical NTK* $\widetilde{\Theta}$ [2] calculated using a finite-width NN: $\widetilde{\Theta}(\mathbf{x}, \mathbf{x}') \triangleq \langle \nabla_\theta f(\mathbf{x}; \theta_0), \nabla_\theta f(\mathbf{x}'; \theta_0) \rangle \approx \Theta(\mathbf{x}, \mathbf{x}')$, where $\theta_0 \sim \mathrm{init}(\cdot)$ denotes initial parameters and $\nabla_\theta f(\mathbf{x}; \theta_0)$ is referred to as the *neural tangent features* [66]. We refer the readers to the works of [2, 27] for a more detailed background on NTK.

**Problem Setting.** We aim to maximize a black-box function $f : \mathcal{X} \to \mathbb{R}$, i.e., find $\mathbf{x}^* \in \arg\max_{\mathbf{x} \in \mathcal{X}} f(\mathbf{x})$, in which the domain $\mathcal{X}$ is a finite subset of the $d$-dimensional unit ball: $\mathcal{X} \subset \{\mathbf{x} | \|\mathbf{x}\|_2 \leq 1\}$. Of note, our theoretical results allow $\mathcal{X}$ to be very large because our regret upper bounds only depend on its cardinality $|\mathcal{X}|$ logarithmically. Moreover, all our theoretical results can be easily extended to problems with continuous input domains with an additional assumption on the Lipschitz continuity of $f$ (Appendix E). We focus on the noisy setting, i.e., for every queried $\mathbf{x}$, we observe a noisy output $y(\mathbf{x}) = f(\mathbf{x}) + \zeta$ where $\zeta \sim \mathcal{N}(0, \sigma^2)$. For simplicity, we focus on the setting of *synchronous* batch BO with a batch size $B$ where a new batch of $B$ inputs are selected only after all evaluations of the previous batch are completed [17]. However, our theoretical results also hold for asynchronous batch BO where a new input query is selected once any pending query is completed (Appendix C). We denote the $i^{\text{th}}$ selected input in iteration $t$ as $\mathbf{x}_t^i$. We analyze the *cumulative regret* of our algorithms: $R_T = \sum_{t=1}^{T/B} \sum_{i=1}^{B} (f(\mathbf{x}^*) - f(\mathbf{x}_t^i))$, because if $R_T$ is shown to be sub-linear in $T$, then the simple regret $S_T = \min_{t,i} (f(\mathbf{x}^*) - f(\mathbf{x}_t^i)) \leq R_T/T$ goes to 0 asymptotically, which implies that our algorithm is *asymptotically no-regret*.

## 3 Sample-Then-Optimize Batch Neural Thompson Sampling

Our STO-BNTS and STO-BNTS-Linear algorithms are presented in Algos. 1 and 2. In both algorithms, the NN surrogate $f(\mathbf{x}; \theta)$ can be either infinite-width or finite-width. Both STO-BNTS and STO-BNTS-Linear follow the TS policy to select an input query $\mathbf{x}_t^i$: They firstly (*a*) obtain a function $f_t^i(\mathbf{x}; \theta_t^i)$ which is equivalently sampled from the GP posterior with the NTK as the kernel: $\mathcal{GP}(\mu_{t-1}(\cdot), \beta_t^2 \sigma_{t-1}^2(\cdot, \cdot))$ [24] (see Appendix A for details), and then (*b*) maximize the function to select the next query: $\mathbf{x}_t^i = \arg\max_{\mathbf{x} \in \mathcal{X}} f_t^i(\mathbf{x}; \theta_t^i)$. Step (*a*) is achieved via the sample-then-optimize procedure, i.e., by firstly *sampling* initial parameters ($\theta_0$ and $\theta_0'$) to construct a function $f_t^i(\mathbf{x}; \theta)$, and then *optimizing* the function using gradient descent to obtain the resulting function of $f_t^i(\mathbf{x}; \theta_t^i)$.

**STO-BNTS (Algo. 1).** In every iteration $t$ of STO-BNTS, we firstly construct an NN $f(\mathbf{x}; \theta)$ and multiply its output by $\beta_t = 2 \log(\pi^2 t^2 |\mathcal{X}| / (3\delta))$, in which $\delta \in (0, 1)$ (Theorem 1).[2] Next, to choose the $i^{\text{th}}$ query $\mathbf{x}_t^i$, we start by sampling initial parameters $\theta_0 \sim \mathrm{init}(\cdot)$ and $\theta_0' \sim \mathrm{init}(\cdot)$ independently, and then set the parameters of $\theta_0'$ in the last layer to 0 (lines 4-5). Next, we use the resulting $\theta_0$ and $\theta_0'$, as well as the NN $f(\mathbf{x}, \theta)$, to construct a function $f_t^i(\mathbf{x}, \theta)$ (line 6). Subsequently, in line 7, using the current history of observations (denoted as $\mathcal{D}_{t-1}$) as the training set, we train $f_t^i(\mathbf{x}, \theta)$ (setting $\theta_0$ as the initial parameters) using gradient descent with the following loss function:

$$\mathcal{L}_t(\theta, \mathcal{D}_{t-1}) = \sum_{\tau=1}^{t-1} \sum_{j=1}^{B} (y_\tau^j - f_t^i(\mathbf{x}_\tau^j; \theta))^2 + \beta_t^2 \sigma^2 \|\theta - \theta_0\|_2^2, \tag{1}$$

in which $\sigma^2$ is the observation noise variance (Sec. 2). After the training, we use the resulting function $f_t^i(\mathbf{x}; \theta_t^i)$ as the *acquisition function* to choose the $i^{\text{th}}$ query: $\mathbf{x}_t^i = \arg\max_{\mathbf{x} \in \mathcal{X}} f_t^i(\mathbf{x}; \theta_t^i)$ (line 8). This procedure (lines 4-8) is repeated independently for $B \geq 1$ times, after which a batch of $B$ queries $\{\mathbf{x}_t^i\}_{i=1,...,B}$ are selected and then queried to produce the observations $\{y_t^i\}_{i=1,...,B}$. Next, the newly collected input-output pairs $\{(\mathbf{x}_t^i, y_t^i)\}_{i=1,...,B}$ are added to $\mathcal{D}_{t-1}$ and the algorithm proceeds to the next iteration. Importantly, if the NN $f(\mathbf{x}; \theta)$ is infinite-width, the function $f_t^i(\mathbf{x}; \theta_t^i)$ obtained after the training in line 7 is *a sample from the GP posterior with the NTK as the kernel*: $\mathcal{GP}(\mu_{t-1}(\cdot), \beta_t^2 \sigma_{t-1}^2(\cdot, \cdot))$ [24] (Appendix A). This ensures the validity of the TS policy and is crucial for deriving the theoretical guarantee of STO-BNTS (Sec. 4).

**STO-BNTS-Linear (Algo. 2).** Similar to STO-BNTS, at the beginning of iteration $t$, STO-BNTS-Linear firstly constructs an NN $f(\mathbf{x}; \theta)$ and multiply its output by $\beta_t$. To choose the $i^{\text{th}}$ query in

---

[2]Note that $\beta_t$ is introduced only for the theoretical analysis, and hence we set $\beta_t = 1$ in our experiments.

---
**Algorithm 1** STO-BNTS
---
1: **for** $t = 1, 2, \dots, T/B$ **do**
2:     Construct NN $f(\mathbf{x}; \theta)$ and multiply its output by $\beta_t$ (Theorem 1)
3:     **for** $i = 1, 2, \dots, B$ **do**
4:        Sample $\theta_0 \sim \text{init}(\cdot)$
5:        Sample $\theta_0' \sim \text{init}(\cdot)$ and set the parameters of $\theta_0'$ in the last layer to 0
6:        Set $f_t^i(\mathbf{x}; \theta) = f(\mathbf{x}; \theta) + \langle \nabla_\theta f(\mathbf{x}; \theta_0), \theta_0' \rangle$
7:        Use observation history $\mathcal{D}_{t-1}$ to train $f_t^i(\mathbf{x}; \theta)$ with the loss function $\mathcal{L}_t(\theta, \mathcal{D}_{t-1})$ (1) (setting $\theta_0$ as the initial parameters) using gradient descent till convergence, to obtain $\theta_t^i = \arg\min_\theta \mathcal{L}_t(\theta, \mathcal{D}_{t-1})$
8:        Choose $\mathbf{x}_t^i = \arg\max_{\mathbf{x} \in \mathcal{X}} f_t^i(\mathbf{x}; \theta_t^i)$
9:     Query the batch $\{\mathbf{x}_t^i\}_{i=1,\dots,B}$ to yield the observations $\{y_t^i\}_{i=1,\dots,B}$, and add them to $\mathcal{D}_{t-1}$

---
**Algorithm 2** STO-BNTS-Linear
---
1: **for** $t = 1, 2, \dots, T/B$ **do**
2:     Construct NN $f(\mathbf{x}; \theta)$ and multiply its output by $\beta_t$ (Theorem 1)
3:     **for** $i = 1, 2, \dots, B$ **do**
4:        Sample $\theta_0' \sim \text{init}(\cdot)$, and define $f_t^i(\mathbf{x}; \theta) = \langle \nabla_\theta f(\mathbf{x}; \theta_0'), \theta \rangle$
5:        Sample $\theta_0 \sim \text{init}(\cdot)$
6:        Use observation history $\mathcal{D}_{t-1}$ to train $f_t^i(\mathbf{x}; \theta)$ with the loss function $\mathcal{L}_t(\theta, \mathcal{D}_{t-1})$ (1) (setting $\theta_0$ as the initial parameters) using gradient descent till convergence, to obtain $\theta_t^i = \arg\min_\theta \mathcal{L}_t(\theta, \mathcal{D}_{t-1})$
7:        Choose the query $\mathbf{x}_t^i = \arg\max_{\mathbf{x} \in \mathcal{X}} f_t^i(\mathbf{x}; \theta_t^i)$
8:     Query the batch $\{\mathbf{x}_t^i\}_{i=1,\dots,B}$ to yield the observations $\{y_t^i\}_{i=1,\dots,B}$, and add them to $\mathcal{D}_{t-1}$

---

iteration $t$, STO-BNTS-Linear firstly constructs a linear model $f_t^i(\mathbf{x}; \theta)$ using the neural tangent features (i.e., $\nabla_\theta f(\mathbf{x}; \theta_0')$ where $\theta_0' \sim \text{init}(\cdot)$ are initial parameters) as the input features (line 4). Next, we sample $\theta_0 \sim \text{init}(\cdot)$ (line 5) and use it as the initial parameters to train $f_t^i(\mathbf{x}; \theta)$ using gradient descent with the same loss function (1) as STO-BNTS, to produce $\theta_t^i$ (line 6). After that, the $i^{\text{th}}$ query is selected by maximizing the acquisition function $f_t^i(\mathbf{x}; \theta_t^i)$: $\mathbf{x}_t^i = \arg\max_{\mathbf{x} \in \mathcal{X}} f_t^i(\mathbf{x}; \theta_t^i)$ (line 7).

Lines 4-6 of STO-BNTS-Linear can be interpreted as a sample-then-optimize method [42] using the neural tangent features as the input features. As a result, same as STO-BNTS, if an infinite-width NN is used, the function $f_t^i(\mathbf{x}; \theta_t^i)$ obtained after the training in line 6 is a sample from the GP posterior with the NTK as the kernel: $\mathcal{GP}(\mu_{t-1}(\cdot), \beta_t^2 \sigma_{t-1}^2(\cdot, \cdot))$ (Appendix A). However, in contrast to STO-BNTS, if the NN is *finite-width*, the function $f_t^i(\mathbf{x}; \theta_t^i)$ derived after line 6 of Algo. 2 still corresponds to a sample from the GP posterior with the *empirical NTK* $\widetilde{\Theta}(\mathbf{x}, \mathbf{x}') = \langle \nabla_\theta f(\mathbf{x}; \theta_0'), \nabla_\theta f(\mathbf{x}'; \theta_0') \rangle$ as the kernel. As a result, for infinite-width NNs, STO-BNTS-Linear and STO-BNTS enjoy the same sub-linear (under some conditions) upper bound on their cumulative regret (Sec. 4.1); however, for finite-width NNs, unlike STO-BNTS, STO-BNTS-Linear still enjoys a regret upper bound which is sub-linear as long as the NN is wide enough (Sec. 4.2).

Although STO-BNTS-Linear (Algo. 2) has the theoretical advantage of a theoretically guaranteed convergence also for finite-width NNs (Sec. 4.2), we expect STO-BNTS (Algo. 1) to perform better in practice. This is because STO-BNTS explicitly trains an NN surrogate model in every iteration and is hence able to *directly exploit the strong representation power of NNs* to model the objective function $f$. In contrast, STO-BNTS-Linear derives its representation power entirely from the neural tangent features of NTK. However, it has been shown [1, 69] that *neural tangent features of NTK can not completely realize the representation power of NNs*. As a result, STO-BNTS is expected to be more competitive in practice, especially in problems where the strong representation power of NNs is essential for accurately modeling the complex objective functions. We empirically verify this practical advantage of STO-BNTS in our experiments (Sec. 5.5).

## 4 Theoretical Results

We firstly prove a regret upper bound for both STO-BNTS and STO-BNTS-Linear using infinite-width NNs, and show that both algorithms are asymptotically no-regret under certain conditions (Sec. 4.1). Next, we derive a regret upper bound for STO-BNTS-Linear when the NN is finite-width, and show that the regret upper bound remains sub-linear as long as the NN is wide enough (Sec. 4.2).

## 4.1 Infinite-width NNs

Here we use $\mathbf{y}_{1:t}$ to denote the output observations from iterations 1 to $t$ (i.e., $t \times B$ observations in total) and use $\mathbf{y}_A$ to denote the vector of observations at a set of inputs $A \subset \mathcal{X}$. Theorem 1 (proof in Appendix C) gives a regret upper bound for both STO-BNTS and STO-BNTS-Linear:

**Theorem 1** (Infinite-width NNs). *Assume that $f$ is sampled from a GP with the NTK $\Theta$ as the kernel function, that $|f(\mathbf{x})| \leq B' \ \forall \mathbf{x} \in \mathcal{X}$ for some $B' > 0$, and that $\Theta(\mathbf{x}, \mathbf{x}') \leq K_0 \ \forall \mathbf{x}, \mathbf{x}' \in \mathcal{X}$ for some $K_0 > 0$. Let $\delta \in (0, 1)$ and $\beta_t = 2 \log(\pi^2 t^2 |\mathcal{X}|/(3\delta))$. Then with probability of $\geq 1 - \delta$,*

$$R_T = \widetilde{\mathcal{O}}(e^C \sqrt{T}(\sqrt{\gamma_T} + 1))$$

*where $\widetilde{\mathcal{O}}$ ignores all logarithmic factors, $\gamma_T$ is the max information gain about $f$ from any set of $T$ observations, and $C$ is an absolute constant s.t. $\max_{A \subset \mathcal{X}, |A| \leq B-1} \mathbb{I}(f; \mathbf{y}_A | \mathbf{y}_{1:t}) \leq C, \forall t \geq 1$.*

The assumption of $\Theta(\mathbf{x}, \mathbf{x}') \leq K_0$ is natural and in line with [31], and $|f(\mathbf{x})| \leq B'$ is also a mild assumption since most practical objective functions have bounded values. Our primary assumption that $f$ is sampled from a GP with the NTK as the kernel function is a common assumption in the analysis of BO algorithms [17, 57]. Of note, compared with the assumptions from previous works using other commonly used kernels such as the squared exponential (SE) kernel [17, 57], our assumption on $f$ holds for more complicated objective functions. Specifically, the class of functions sampled from a GP with the NTK as the kernel subsumes more non-smooth and sophisticated objective functions compared with the other commonly used kernels. As shown in [17], $C$ can be chosen to be an absolute constant that is independent of $B$ and $T$ as long as we initialize our algorithm using the uncertainty sampling method, which entails choosing the initial inputs by sequentially maximizing the GP posterior variance (more details in Appendix B). Specifically, for any chosen constant $C > 0$, as long as we run the uncertainty sampling initialization stage for $T_{\text{init}}$ iterations such that $((B-1)\gamma_{T_{\text{init}}})/T_{\text{init}} \leq C$, then $\max_{A \subset \mathcal{X}, |A| \leq B-1} \mathbb{I}(f; \mathbf{y}_A | \mathbf{y}_{1:t}) \leq C, \forall t \geq 1$ is guaranteed to be satisfied. Since it has been shown by the work of [31] that $\gamma_T = \widetilde{\mathcal{O}}(T^{(d-1)/d})$ grows sub-linearly for the NTK,[3] therefore, $((B-1)\gamma_{T_{\text{init}}})/T_{\text{init}}$ is decreasing as $T_{\text{init}}$ increases. As a result, for any chosen $C$, we are able to choose a finite $T_{\text{init}}$ such that the condition $((B-1)\gamma_{T_{\text{init}}})/T_{\text{init}} \leq C$ is satisfied.

For example, if we choose $C = 1$, then the required number of initial iterations is approximately $T_{\text{init}} = \Theta((B-1)^d)$. As a result, the regret upper bound will be a summation of $2B'T_{\text{init}}$ (i.e., the regrets incurred during the initializatioin stage, because the regret at every step is upper-bounded by $2B'$) and the regret upper bound from Theorem 1 with $C = 1$. Since both $B'$ and $T_{\text{init}}$ are constants independent of $T$ (assuming that $B$ is independent of $T$), the asymptotic regret upper bound can be simplified into $R_T = \widetilde{\mathcal{O}}(\sqrt{T}(1 + \sqrt{\gamma_T}))$. Plugging in $\gamma_T = \widetilde{\mathcal{O}}(T^{(d-1)/d})$, the final regret upper bound becomes $R_T = \widetilde{\mathcal{O}}(T^{1/2} + T^{(2d-1)/(2d)}) = \widetilde{\mathcal{O}}(T^{(2d-1)/(2d)})$ which is sub-linear in $T$ and hence implies that our STO-BNTS and STO-BNTS-Linear are both *asymptotically no-regret* when the NN surrogate is infinite-width.

Moreover, when $T \gg B$ and $B$ is a constant which is independent of $T$, Theorem 1 gives us insights on the benefit of batch over sequential evaluations. In this case, the regrets incurred during initialization (i.e., $\widetilde{\mathcal{O}}((B-1)^d)$) is negligible. Therefore, our analysis above suggests that our algorithms with batch evaluations ($B > 1$) enjoys the same asymptotic regret upper bound as its sequential counterpart ($B = 1$) since the resulting regret upper bound of $R_T = \widetilde{\mathcal{O}}(\sqrt{T}(1 + \sqrt{\gamma_T}))$ does not depend on the batch size $B$. This demonstrates the advantage of batch evaluations because when $B > 1$, some of our evaluations can be run in parallel, which is not supported by the sequential setting with $B = 1$. As a simple illustration, for a large $T$, both the sequential and batch settings achieve a simple regret of the order $\widetilde{\mathcal{O}}(T^{-1/(2d)})$ after $T$ function evaluations. However, since our batch setting can evaluate every $B > 1$ selected inputs in parallel in every iteration, our batch setting with $B > 1$ achieves this simple regret after only $T/B$ iterations, which is only a fraction $(1/B)$ of the $T$ iterations required by the sequential setting. This also shows that we enjoy more benefit with a larger batch size $B$.

---

[3]Note that in order to quote the results from [31], we need follow their assumption to assume that the domain $\mathcal{X}$ is a subset of the $d$-dimensional unit hyper-sphere: $\mathcal{X} \subset \{\mathbf{x} | \|\mathbf{x}\|_2 = 1\}$, which is more strict than our main assumption (Sec. 2) that $\mathcal{X}$ is a subset of the unit hyper-ball: $\mathcal{X} \subset \{\mathbf{x} | \|\mathbf{x}\|_2 \leq 1\}$.

## 4.2 Finite-width NNs

Here we prove a regret upper bound for STO-BNTS-Linear with *finite-width* NN. The main technical challenge in the proof lies in the disparity between the exact and empirical NTKs, i.e., the function $f$ is assumed to be sampled from a GP with the exact NTK $\Theta$ yet our acquisition function $f_t^i(\mathbf{x}; \theta_t^i)$ (line 7 of Algo. 2) is obtained using the empirical NTK $\widetilde{\Theta}$ when the NN is finite-width. To this end, we make use of the following theoretical guarantee on the approximation error between $\Theta$ and $\widetilde{\Theta}$.

**Proposition 1** (Theorem 3.1 of [2]). *Choose $\varepsilon > 0$ and $\delta \in (0, 1)$. If the width of every layer in an NN satisfies $m = \Omega(\frac{L^6}{\varepsilon^4} \log(4L|\mathcal{X}|^2/\delta))$, then $\forall \mathbf{x}, \mathbf{x}' \in \mathcal{X}$, we have with probability $\geq 1 - \delta/4$ that*

$$\left| \langle \nabla_\theta f(\mathbf{x}, \theta_0), \nabla_\theta f(\mathbf{x}', \theta_0) \rangle - \Theta(\mathbf{x}, \mathbf{x}') \right| \leq (L+1)\varepsilon.$$

Proposition 1 ensures that we can reduce the upper bound $(L+1)\varepsilon$ on the approximation error between $\Theta$ and $\widetilde{\Theta}$ by increasing the width $m$ of the NN. For example, let $m = C_{\text{ntk}} L^6 \varepsilon^{-4} \log(4L|\mathcal{X}|^2/\delta)$ for some constant $C_{\text{ntk}} > 0$, then the approximation quality of Proposition 1 can be expressed as $(L+1)\varepsilon = C_{\text{ntk}}(L+1)L^{3/2} \log^{1/4}(4L|\mathcal{X}|^2/\delta)m^{-1/4}$ which decreases as the width $m$ increases. In our theoretical analysis, we assume that $(L+1)\varepsilon \leq 1$, which can be satisfied as long as $m$ is large enough. The regret upper bound for STO-BNTS-Linear is given in Theorem 2 (proof in Appendix D):

**Theorem 2** (Finite-width NNs). *Let $\delta \in (0, 1)$ and $\beta_t = 2 \log(2\pi^2 t^2 |\mathcal{X}|/(3\delta))$. Then we have*

$$R_T = \widetilde{\mathcal{O}}(e^C \sqrt{T}(\sqrt{\gamma_T} + 1) + T^3 m^{-1/8}(L+1)^{5/4}),$$

*with probability of $\geq 1 - \delta$. Here $\gamma_T$ and $C$ are the same as those defined in Theorem 1.*

The first term in the regret upper bound in Theorem 2 is the same as that of Theorem 1 for infinite-width NNs and can hence be made sub-linear by using uncertainty sampling as the initialization stage: $R_T = \widetilde{\mathcal{O}}(\sqrt{T}(1 + \sqrt{\gamma_T})) = \widetilde{\mathcal{O}}(T^{(2d-1)/(2d)})$. The second term in the regret upper bound represents the additional regrets incurred by the use of finite-width NNs, which can also be made sub-linear by choosing $m = \Omega(T^{24})$. In other words, if the width $m$ of the NN is chosen to be large enough (i.e., if $m = \Omega(T^{24})$), then the cumulative regret of STO-BNTS-Linear scales sub-linearly in $T$.

## 4.3 Discussion

The assumption on the function $f$ for our theoretical analyses in Theorems 1 and 2 differs from those made by the previous works on neural contextual bandits [66, 69]. Specifically, these previous works have relied on the assumption of a positive definite NTK Gram matrix to approximate the value of $f$ (only evaluated at the observed contexts up to iteration $T$) using a function that is linear in the neural tangent features. When translated into our setting (which is equivalent to the contextual bandit setting where all contexts are fixed for all $t \geq 1$), the assumption of a positive definite NTK Gram matrix can be easily violated as long as any input $\mathbf{x}$ is queried more than once, thus rendering this assumption unrealistic in our setting. In contrast, we have assumed that $f$ is sampled from a GP with the NTK as the kernel, which is a common assumption in the analysis of BO [17, 57] and allows us to derive a sub-linear regret upper bound. As a result of the different assumptions, our regret upper bounds are not directly comparable with those from the previous works on neural contextual bandits [66, 69].

# 5 Experiments

We compare our STO-BNTS and STO-BNTS-Linear with the baselines of Neural UCB [69] and Neural TS [66], as well as GP-UCB and GP-TS which use GPs (with the SE kernel) instead of NNs as their surrogate models. The original implementations of Neural UCB [69] and Neural TS [66] are only applicable to discrete domains. So, for a fair comparison in those tasks with continuous domains, we have modified their implementations to maximize their acquisition functions in the same way as our methods (i.e., through a combination of random search and L-BFGS-B, refer to Appendix F for more details). We firstly explore some interesting insights about our algorithms using a synthetic experiment in Sec. 5.1. Next, we apply our algorithms to real-world AutoML (Sec. 5.2) and RL (Sec. 5.3) problems, as well as an optimization task over images (Sec. 5.4). Finally, we discuss some interesting insights from our experiments in Sec. 5.5. We plot the simple regret (or the best found observation till an iteration) when presenting the experimental results, which is the common practice in BO [8, 29]. We have deferred some experimental details to Appendix F due to space limitation. Our code is available at `https://github.com/daizhongxiang/sto-bnts`.

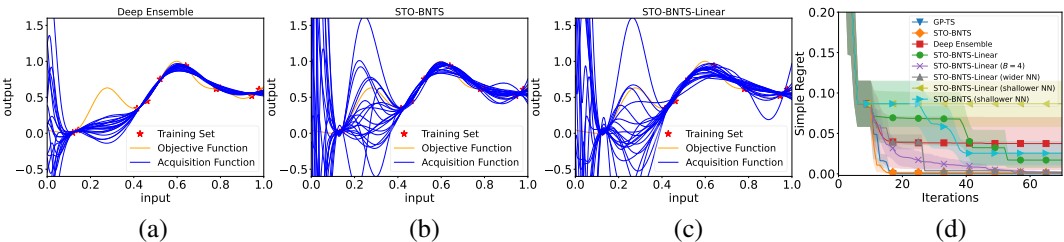

Figure 1: (a-c) Acquisition functions and (d) performances in the synthetic experiment (Sec. 5.1).

## 5.1 Synthetic Experiment

Here, we sample a smooth function from a GP with an SE kernel (with a length scale of $0.1$), defined on a discrete 1-dimensional domain within the range of $[0, 1]$. For all methods, we use an NN architecture with a depth of $L = 8$ and a width of $m = 64$ unless specified otherwise.

Figs. 1 (a-c) illustrate the acquisition functions $f_t^i(\mathbf{x}; \theta_t^i)$ (line 8 of Algo. 1 and line 7 of Algo. 2) of different algorithms. The *Deep Ensemble* method [36] in Fig. 1a can be regarded as a reduced version of our STO-BNTS algorithm (Algo. 1) in which the term $\langle \nabla_\theta f(\mathbf{x}; \theta_0), \theta_0' \rangle$ (i.e., the second term in line 6 of Algo. 1) is removed. As a result, Deep Ensemble does not enjoy the theoretical guarantees of our algorithms (Sec. 4). In Figs. 1 (a-c), given the same training set (red stars), every method constructs a batch of $B = 100$ acquisition functions. E.g., STO-BNTS repeats lines 4-7 of Algo. 1 independently for $B = 100$ times to produce acquisition functions $f_t^i(\mathbf{x}; \theta_t^i)$ for $i = 1, \ldots, 100$, which are plotted as the blue lines in Fig. 1b. Note that every acquisition function (blue line) is maximized to select an input query. The figures show that compared with the naive baseline of Deep Ensemble, our STO-BNTS and STO-BNTS-Linear are able to display more exploratory behaviors in unexplored regions (e.g., the interval of $[0.2, 0.4]$). This may be explained by our theoretical guarantees (Sec. 4) which imply that both of our algorithms are able to perform exploration in a principled way and hence naturally handle the the exploration-exploitation trade-off. Moreover, it has also been justified by [24] that the addition of the term $\langle \nabla_\theta f(\mathbf{x}; \theta_0), \theta_0' \rangle$ improves the ability of the NN to characterize the uncertainty of predictions, which corroborates our findings here.

The simple regrets of different algorithms are plotted in Fig. 1d.[4] The first interesting observation is that our STO-BNTS (orange) significantly outperforms Deep Ensemble (red), which corroborates the insight discussed above positing that the addition of the term $\langle \nabla_\theta f(\mathbf{x}; \theta_0), \theta_0' \rangle$ leads to more principled exploration and hence better performances. Due to its lack of exploration as illustrated in Fig. 1a, Deep Ensemble fails to reach zero regret in Fig. 1d. Furthermore, the discrepancy between the green and purple curves shows that batch evaluations ($B = 4$) lead to significant performance improvement. Moreover, compared with the green curve for which $m = 64$, *using a wider NN* (gray curve, $m = 512$) *substantially improves the performance of STO-BNTS-Linear* yet employing a shallower NN (yellow curve, $m = 16$) significantly degrades the performance. These observations agree with Theorem 2 which states that a larger width $m$ reduces the regret of STO-BNTS-Linear. Similarly, the NN surrogate model of STO-BNTS should also be wide enough since the use of a narrower NN (light blue curve, $m = 16$) also leads to a worse performance for STO-BNTS. Lastly, GP-TS (blue) performs competitively in this experiment with a very smooth objective function. However, as we will show in the next two sections, in real-world experiments with more complicated objective functions, GP-based methods are consistently outperformed by our algorithms.

## 5.2 Real-world Experiments on Automated Machine Learning (AutoML)

Here, we adopt 3 hyperparameter tuning tasks. We use tasks involving categorical hyperparameters to highlight the advantage of our NN-based over GP-based methods. We use a diabetes diagnosis dataset to tune 6 categorical hyperparameters of random forest (RF), and then use the MNIST dataset to tune 9 hyperparameters (4 continuous and 5 categorical) of XGBoost and 9 hyperparameters (2 continuous and 7 categorical) of convolutional neural networks (CNNs). Fig. 2 plots the results for the RF (a,b) and XGBoost (c,d) tasks, and the results for CNN are shown in Fig. 4 (Appendix F.2).

---

[4]To show the benefit of batch evaluations, in all experiments (including real-world experiments), we use the *iterations* $t$ as the horizontal axis and in every iteration $t$, we report the largest $f(\mathbf{x}_t^i)$ ($y_t^i$ in real-world experiments) within a batch (when $B > 1$) as the observation in this iteration.

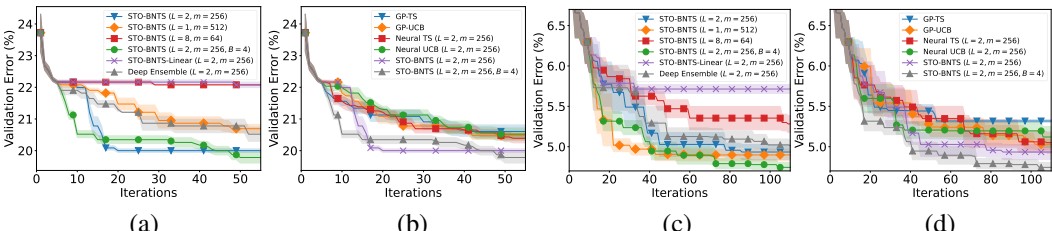

Figure 2: Validation errors for hyperparameter tuning of RF (a,b), XGBoost (c,d). $B = 1$ by default.

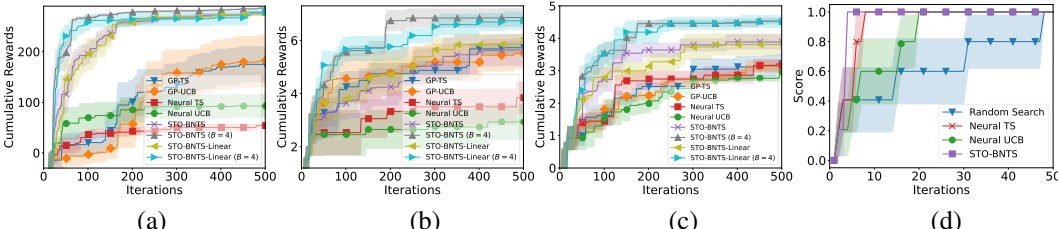

Figure 3: Results for (a) 12-D Lunar-Lander, (b) 14-D robot pushing, (c) 20-D rover trajectory planning, and (d) optimization over images in Sec. 5.4. $B = 1$ unless specified otherwise.

For each task, we firstly compare the performances of different variants of our algorithms in Figs. 2a, c and Fig. 4a, which demonstrate a number of interesting insights. Firstly, our STO-BNTS outperforms Deep Ensemble, which is consistent with the results in Fig. 1d and further emphasizes the practical significance of the improved exploration performed by our STO-BNTS (Sec. 5.1). Secondly, STO-BNTS-Linear is significantly outperformed by STO-BNTS in Figs. 2 a and c, which can likely be attributed to its inability to fully leverage the representation power of NNs as we have discussed in Sec. 3 (see more discussions in Sec. 5.5). Next, Figs. 2a, c and Fig. 4a also show that the performance of STO-BNTS tends to suffer when the NN surrogate model is either overly shallow ($L = 1, m = 512$, orange curves) or overly deep ($L = 8, m = 64$, red curves), that is, both the orange and red curves underperform significantly in two of the three experiments. This is likely because an overly shallow NN lacks the representation power to model complicated objective functions, whereas an overly deep NN may be prone to overfitting since the size of the training set here is much smaller than typical deep learning applications. In contrast, the NN architecture of $L = 2, m = 256$ (blue curves) consistently performs well in all three experiments, therefore, we will use it as the default architecture in the experiments in the next section. Moreover, the benefit of batch evaluations can also be further confirmed by comparing the blue ($B = 1$) and green ($B = 4$) curves in Figs. 2a, c and Fig. 4a.

Figs. 2b, d and Fig. 4b show that for the same NN architecture of $L = 2, m = 256$, our STO-BNTS is the best-performing method since it performs better than all baselines in Figs. 2b and d and comparably with them in Fig. 4b. The undesirable performances of Neural UCB and Neural TS may be explained by the errors due to their diagonal matrix approximation (Sec. 1), and the underwhelming performances of GP-UCB and GP-TS may result from the ineffectiveness of GP in modeling categorical inputs (more on this in Sec. 5.5).

## 5.3 Real-world Experiments on Reinforcement Learning (RL)

Here we optimize the control parameters of 3 RL problems: we tune $d = 12$ parameters of a heuristic controller for the Lunar-Lander task from OpenAI Gym [4], $d = 14$ parameters of a controller for a robot pushing task [62], and $d = 20$ parameters for a rover trajectory planning problem [62]. We use $L = 2, m = 256$ for all methods. The results in these three RL tasks are plotted in Figs. 3a, b and c. Our STO-BNTS and STO-BNTS-Linear (purple and yellow curves) consistently perform the best among all methods with sequential evaluations ($B = 1$), and our methods with batch evaluations ($B = 4$, gray and light blue curves) achieve further performance improvements over their sequential counterparts. Of note, despite underperforming in Sec. 5.2, STO-BNTS-Linear achieves comparable performances with STO-BNTS in all three experiments here for both the sequential and batch evaluations, outperforming the other baselines. The inefficacy of GP-TS and GP-UCB here may result from the inability of GPs to effectively model high-dimensional input space [30] (Sec. 5.5).

## 5.4 Optimization over Images

Some real-world applications require optimizing over an input domain of *images*. For example, an image recommender system sequentially recommends different images to a user in order to select the image with the best rating/score [65]. In these applications, GP-based BO methods require sophisticated techniques such as convolutional kernels in order to model the inputs of images [59], in contrast, our methods can be easily applied by simply replacing the NN surrogate model with a CNN. Here, we simulate these applications by maximizing a score over the domain of images from the MNIST dataset. For all CNN-based methods, we use a CNN with a convolutional layer followed by a fully connected layer (both with $m = 64$) as the surrogate model (more details in Appendix F.4). The results (Fig. 3d) show that our STO-BNTS performs the best. Moreover, when using the same CNN architecture, STO-BNTS-Linear fails to achieve comparable performances to the other CNN-based methods since STO-BNTS-Linear is not able to fully leverage the representation power of CNN (Sec. 3). However, again consistent with Theorem 2, increasing the width of the CNN can improve the performance of STO-BNTS-Linear to be comparable with STO-BNTS (Fig. 6 in Appendix F.4).

## 5.5 Discussion

Secs. 5.2, 5.3 and 5.4 show that our algorithms, although *without any special design for a specific type of problem* (e.g., problems involving categorical, high-dimensional or image inputs), are *competitive in all these problems* thanks to the ability of NNs to model complicated real-world functions. Compared with GP-based BO methods, our algorithms may incur more computation due to the need to train an NN surrogate model. However, the additional computation can be easily overshadowed by the cost of function evaluations since BO is usually used to optimize expensive-to-compute functions.

**STO-BNTS vs. STO-BNTS-Linear.** Our experimental results have demonstrated some interesting insights on the comparison between our STO-BNTS (Algo. 1) and STO-BNTS-Linear (Algo. 2). As we have discussed in Sec. 3, since STO-BNTS-Linear is unable to explicitly take advantage of the representation power of NNs, it is expected to be less competitive than STO-BNTS in practice especially in problems where the strong representation power of NNs is crucial for accurately modeling the objective function. Examples of such problems include those with categorical (Sec. 5.2) or image (Sec. 5.4) inputs. Interestingly, our results in Secs. 5.2 and 5.4 indeed corroborate this insight by showing that STO-BNTS-Linear is consistently outperformed by STO-BNTS in these experiments. However, note that in such problems, the performance of STO-BNTS-Linear can be significantly improved by further increasing the width of the NN (CNN) as we have shown in Fig. 6. In addition, the practical efficacy of STO-BNTS-Linear can also be seen from the experiments in Sec. 5.3, in which STO-BNTS-Linear performs comparably with STO-BNTS and *consistently outperforms all other baselines*. This demonstrates the empirical competence of STO-BNTS-Linear in some real-world problems such as RL. Moreover, also note that compared with STO-BNTS, STO-BNTS-Linear enjoys the theoretical advantage of having a guaranteed convergence for finite-width NNs (Sec. 4.2).

**Baseline Methods.** The underwhelming performances of Neural UCB and Neural TS in our experiments are likely caused by the errors introduced by the diagonal matrix approximation that is used to avoid the inversion of the $p \times p$ matrix (Sec. 1). The inadequate performances of GP-UCB and GP-TS may be explained by the ineffectiveness of GPs to model objective functions involving categorical [15, 18] or high-dimensional inputs [30, 43], which correspond to the experiments in Secs. 5.2 and 5.3, respectively. Nevertheless, we expect GP-based methods to achieve better performances (than their performances here) in problems with lower-dimensional and purely continuous input space (e.g., GP-TS performs competitively in the synthetic experiment as shown in Fig. 1d).

**Depth $L$ and Width $m$ of the NN Surrogate Model.** Our experimental results have provided some guidelines on the choices of the depth $L$ and width $m$ of the NN surrogate model in our algorithms. Regarding the depth $L$, our experiments in Sec. 5.2 have shown that an overly shallow NN usually hurts the performance due to its lack of expressive power, whereas an excessively deep NN is also likely to deteriorate the performance due to overfitting. Therefore, we discourage the use of NNs which are either exceedingly shallow or overly deep, and recommend shallower NNs for simpler tasks to prevent overfitting and deeper NNs for more complicated tasks to gain enough representation power. The width $m$ should be chosen to be large enough since our experiments in Sec. 5.1 (Fig. 1d) and Sec. 5.4 (Fig. 6 in Appendix F.4) suggest that a larger width usually improves the performance. Moreover, we have shown that the choice of $L = 2, m = 256$ consistently leads to competitive performances in a wide range of experiments (i.e., all experiments in Sec. 5.2 and Sec. 5.3). Therefore, we recommend it as the default choice for real-world problems.

# 6    Related Works

The NTK provides a theoretical tool to study the training dynamics of NNs by drawing connections with kernel methods [2, 5, 13, 27, 38], and it has been successfully applied to a number of practical problems such as neural architecture search [52, 54], active learning [61], data valuation [63], among others. The pioneering work of [69] exploited the theory of NTK to introduce the Neural UCB algorithm for contextual bandits, which uses an NN to learn the reward (objective) function and leverages neural tangent features for exploration following the UCB principle. Neural UCB has been followed up by other works which mainly aimed to make Neural UCB more practical. [64] proposed to improve the computational efficiency of Neural UCB through the additional assumption that the reward function is linear in the feature mapping of the last layer of the NN, [23] also aimed to reduce the computational cost of Neural UCB by limiting the number of updates of the NN surrogate model, and [44] proposed a method to reduce the memory requirement of generic neural bandit algorithms. The recent work of [41] performed an empirical study of neural bandit algorithms, and discovered that they tend to perform competitively in problems that require learning complicated representations. Above all, the work that is most closely related to our paper is [66], which introduced the Neural TS algorithm. Following similar principles as Neural UCB, Neural TS learns the reward function using an NN surrogate model and constructs the exploration term through the neural tangent features, which are used as, respectively, the mean and variance of the Gaussian distribution from which the reward is sampled for running the TS routine. GP-based BO methods [3, 33, 48, 49] have achieved impressive performances in recent years, and have been extended to various problem settings such as high-dimensional BO [19, 26, 30], multi-fidelity BO [14, 28, 67, 68], meta-BO [10, 50], risk-averse BO [46, 47, 58], multi-agent/collaborative BO [9, 56], non-myopic BO [34, 40], among others. More importantly, they have also been extended to the batch setting, based on either GP-UCB [8, 16, 17, 32, 60] or GP-TS [25, 29, 45, 60].

# 7    Conclusion

We propose STO-BNTS and STO-BNTS-Linear, both of which sidestep the requirement to invert a large parameter matrix in existing neural bandit algorithms, and naturally support batch evaluations while preserving their theoretical guarantees. Both algorithms are asymptotically no-regret under certain conditions if the NN surrogate is infinite-width, and STO-BNTS-Linear still enjoys sub-linear regret for a finite-width NN if it is wide enough. A potential limitation is that our theoretical analysis for finite-width NNs (Sec. 4.2) only holds for STO-BNTS-Linear but not for STO-BNTS, which we will explore in future works. Another promising future topic is to leverage our ability to exploit the strong representation power of NNs to apply our algorithms to other challenging optimization tasks with sophisticated search spaces, such as chemical design [22, 35], neural architecture search [53, 55], etc. Moreover, it is also an interesting future direction to extend our algorithms to handle other problem settings such as those which have been considered by BO (Sec. 6), e.g., multi-fidelity optimization [67, 68], risk aversion/fault tolerance [20, 46, 47], etc. A potential negative societal impact is that our work may promote more adoption of deep learning methods and hence cause more electricity consumption.

## Acknowledgments and Disclosure of Funding

This research/project is supported by A*STAR under its RIE2020 Advanced Manufacturing and Engineering (AME) Industry Alignment Fund – Pre Positioning (IAF-PP) (Award A19E4a0101).

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
