# OpenReview forum: "Sample-Then-Optimize Batch Neural Thompson Sampling"
_NeurIPS.cc/2022/Conference — NeurIPS 2022 Accept_

### Official Review · Reviewer_an5h · 2022-07-10

**Rating:** 6
**Confidence:** 4
**Soundness:** 3 good
**Presentation:** 4 excellent
**Contribution:** 3 good

**Summary:**

The authors propose two kinds of new Bayesian Optimization (BO) methods in this work. They use the neural tangent kernel Gaussian Process (NTKGP) as the surrogate model and do the neural Thompson Sampling by sampling the initialization of the neural network and optimizing it to get the next query. The major effort of this work is to prove the sublinear regret of their methods, and the experimental results seem promising.

**Questions:**

I have the following questions/comments. I am willing to adjust my score further if the feedbacks from the authors make sense to me.

Theorem 1

(a) The proof is based on finite inputs, which still leaves a gap between theory and practice. What is the major obstacle to generalizing this result to the input of the continuous space?

(b) The authors may consider being more precise when discussing the batch size $B$. It is not completely correct that the regret upper bound does not depend on $B$. The constant $C$ is related to $B$. As the authors say in the manuscript, If we want that maximal conditional mutual information to be no larger than $C$, then we cannot set $B$ to a very high value, otherwise the number of initialized points needs to be extremely high. The authors say that when $C=1$, then $T_{init}=\Theta ((B-1)^{d})$, meaning that even $B=3$ and $d=10$ will make the number of initialized points unaffordable to almost all the BO cases. The last paragraph of section 4.1 will give people a misconception that it will be totally safe however large $B$ is chosen, but think of an extreme case: when $B=T$, I don't think the sublinear regret bound will still hold.

(c) The authors use a result from previous work to bound $\gamma_{T}$, but as they point out, the input domain condition of that previous work is different from this work. Why do the authors think it is reasonable to borrow that previous result?

(d) Lemma 4, $p-1/t^2<0$ when $t< 5$, making the statement trivial. Also, when $t<5$, it is not true that $1/(p-1/t^2)\leq 10/p$ (proof of lemma 6). The authors may consider discussing the value of $t$ seperately when $t<5$ and when $t \geq 5$ (although I don’t think it will affect the final theoretical result).

(e) Proof of lemma 7, is $Y_t$ always integrable?

(f) Minor point: math form (13), (14), some “)” are redundant.

Theorem 2

(a) It is not convincing to me to have the claim “STO-BNTS-Linear remains asymptotically no-regret as long as the NN is wide enough” based on the result of Theorem 2. That claim means that $\exists N>0$, s.t, $\forall m>N$ ($m$ is the width of NN), $\lim_{T\to\infty}\frac{R_{T}}{T}= 0$ when $m$ is fixed. $N$ can be very large but $m$ should be considered a constant. The result of Theorem 2 gives us some intuition on how to choose $m$ when we know the value of $T$ (but even so, $\Omega(T^{24})$ is a very loose bound), but it can not give the asymptotically no-regret conclusion for the finite-width NNs.

(b) Math form (26), should the first $=$ be $\leq$? (the operator norm is a sub-multiplicative matrix norm)

(c) Math form (27), the third $\leq$ from last is $=$.

(d) Math form (28), the first $\leq$ is $=$.

(e) Math form (41), the $\leq$ between $(a)\leq$ and $(b)\leq$ is $=$.

Experiments

(a) How to set the value of $\sigma$ in the loss function?

(b) The authors say that they use L-BFGS-B to optimize NN and get the new query, how does this work when the input domain is discrete/categorical or mixed?

(c) The authors say that they also use L-BFGS-B to optimize the acquisition function of Neural Thompson Sampling. What kind of acquisition function is used here?

(d) What kind of implementation do the authors use for GP-TS and GP-UCB? From my previous experiences, I find that even for the same algorithm different implementations will make a large difference in the results. It makes sense to me that standard GP-UCB and GP-TS cannot deal with categorical inputs well, but I am a little bit surprised that they perform so badly in cases with continuous spaces (Figure 3(a)-(c)). After all, according to the recent developments of high-dimensional BO $d\leq 20$ is actually not so high-dimensional.

(e) It might not be so fair to compare sequential baselines with STO-BNTS ($B>1$) under fixed iterations. In that case, STO-BNTS ($B>1$) evaluates the black-box function $B$ times as other methods. It would be more reasonable to compare batched STO-BNTS with other batched baselines such as batched GP-TS (under the same batch size).




**Limitations:**

The authors mention the limitations and potential negative societal impact of their work in the Conclusion section of their manuscript. The authors may consider discussing the limitations of their work in more detail based on the Questions listed above. I don't think this work will have any severe negative societal impact.

**Strengths And Weaknesses:**

Currently, my overall evaluation of this work is positive. Although NTKGP has already been published before, this is the first work to use it in Bayesian Optimization. I think this work provides a new Bayesian Optimization framework that uses a non-standard Gaussian Process as the surrogate and has a theoretical guarantee. The experimental results shown in the manuscript indicate that this new framework has the potential to outperform standard Bayesian Optimization methods in many black-box optimization cases. The proof of theorems seems non-trivial.

I have a couple of questions (including some weaknesses from my perspective) based on the manuscript, and they are listed in the Questions section.

---

> ### Author Response · Authors · 2022-08-02
> **Response to Reviewer an5h (Part 1/3)**
>
>
> We would like to thank the reviewer for your detailed and insightful comments, which we will follow seriously to improve our paper.
>
> ---
>
> # Theorem 1
>
> > (a) The proof is based on finite inputs, which still leaves a gap between theory and practice. What is the major obstacle to generalizing this result to the input of the continuous space?
>
> Generalizing our theoretical results to continuous input domains can be achieved by adding an additional assumption on the Lipschitz continuity of the objective function $f$ (e.g., a similar assumption to the one used by Theorem 2 of [30]).
> With this additional assumption, our theoretical results can be extended to continuous domains by following existing techniques.
> For example, we can simply construct a finite sub-domain of the continuous domain (with small equal spacing in each dimension) and then run our algorithm on this finite sub-domain.
> As a result, compared with our regret upper bounds for finite domains, this only incurs an additional multiplicative factor of $\mathcal{O}(\log T)$ and an additional additive factor of $\mathcal{O}(\sqrt{T})$ which does not affect the scaling of the regret upper bound in $T$ (refer to Section 3.1 of paper [a] below for more details).
>
> We did not use this assumption to generalize our theoretical results to continuous domains because although this assumption has been shown to hold for commonly used kernels such as the squared exponential kernel (Theorem 2 of [30]), to the best of our knowledge, it has not been shown to hold for the NTK. We will add our discussions here to the paper.
>
> [a] Gaussian Process Bandit Optimization with Few Batches, AISTATS 2022.
>
> ---
>
> > (b) The authors may consider being more precise when discussing the batch size $B$. It is not completely correct that the regret upper bound does not depend on $B$. The constant $C$ is related to $B$. As the authors say in the manuscript, If we want that maximal conditional mutual information to be no larger than $C$, then we cannot set $B$ to a very high value, otherwise the number of initialized points needs to be extremely high. The authors say that when $C=1$, then $T_{\text{init}}=\Theta((B-1)^d)$, meaning that even $B=3$ and $d=10$ will make the number of initialized points unaffordable to almost all the BO cases. The last paragraph of section 4.1 will give people a misconception that it will be totally safe however large $B$ is chosen, but think of an extreme case: when $B=T$, I don't think the sublinear regret bound will still hold.
>
> You are correct that our statement of "our asymptotic regret upper bound does not depend on $B$" (lines 193-195) needs certain conditions in order to hold.
> To clarify, this statement is in fact only applicable when $T$ is large (as we mentioned in line 197) compared with $B$ and $B$ is a constant which is independent of $T$. It is no longer applicable if either $B$ is large relative to $T$ or if $B$ grows with $T$ (e.g., the extreme case you have mentioned when $B=T$). The reason is exactly as you have mentioned: If $B$ is large relative to $T$ or $B$ grows with $T$, then the regrets during initialization $\mathcal{O}((B-1)^d)$ will no longer be negligible. Therefore, we can no longer simplify the regret upper bound into $R_T=\widetilde{\mathcal{O}}(\sqrt{T}(1+\sqrt{\gamma_T}))$ (lines 187-189). As a result, the subsequent statement in the next paragraph (which is based on this simplified regret upper bound) saying that "the asymptotic regret upper bound does not depend on $B$" will no longer hold.
> Similarly, when saying that our regret upper bound is sub-linear in $T$ (line 190), we are assuming that the batch size $B$ is a constant which is independent of $T$ so that the regrets during initialization $\mathcal{O}((B-1)^d)$ do not affect the scaling in $T$.
>
> Thank you pointing out this confusion, we will revise this section by incorporating our discussions here to make our statements more accurate.
>
> ---
>
> &#8595; &#8595; &#8595; **Continued below** &#8595; &#8595; &#8595;

---

> > ### Author Response · Authors · 2022-08-02
> > **Response to Reviewer an5h (Part 2/3)**
> >
> >
> >
> > # Theorem 1 (Cont'd)
> >
> > > (c) The authors use a result from previous work to bound $\gamma_T$, but as they point out, the input domain condition of that previous work is different from this work. Why do the authors think it is reasonable to borrow that previous result?
> >
> > Our assumption on the input domain $\mathcal{X}$ is in fact more general than that from the previous work of [19].
> > Specifically, [19] assumes that $\mathcal{X} \subset \{\mathbf{x}|||\mathbf{x}||_2=1\}$.
> > In contrast, our entire regret analyses (to derive Theorem 1 and Theorem 2) are based on the more general assumption of $\mathcal{X} \subset \{\mathbf{x}|||\mathbf{x}||_2\leq 1\}$,
> > and only the additional results (after our regret analysis) which make use of the growth rate of $\gamma_T$ from [19] require the more strict assumption from [19].
> > In other words, we can also choose to adopt the more strict assumption from [19] from the beginning, and all our theoretical results will still hold.
> > In this case, this more strict assumption on $\mathcal{X}$ from [19] is in fact the same as most previous works on neural bandits such as Neural UCB [36] and Neural TS [35].
> > We will add these discussions to the paper to avoid potential confusions.
> >
> > ---
> >
> > > (d) Lemma 4, $p-1/t^2 < 0$ when $t< 5$, making the statement trivial. Also, when $t< 5$, it is not true that $1/(p-1/t^2) \leq 10 / p$ (proof of lemma 6). The authors may consider discussing the value of $t$ separately when $t < 5$ and when $t\geq 5$ (although I don’t think it will affect the final theoretical result).
> >
> > You are correct that when $t < 5$, we have that $p-1/t^2 < 0$ in Lemma 4. However, even when $t < 5$, the proof of Lemma 6 still holds because $1/(p-1/t^2) \leq 10 / p$ is still valid (since the left hand side is $ < 0$ and the right hand side is $> 0$).
> > We will add these details to avoid potential confusions.
> >
> > ---
> >
> > > (e) Proof of lemma 7, is $Y_t$ always integrable?
> >
> > Yes, $Y\_t=\sum^t_{s=1}X\_s$ is always integrable. To begin with, note that conditioned on $\mathcal{F}\_{t-1}$, we have that $X_1,\ldots,X_{t-1}$ are deterministic and only $X_t$ is random whose randomness results from $\mathbf{x}\_t$.
> > So, the conditional expectation in equation (19) in the proof of Lemma 7 is taken over the randomness of $\mathbf{x}\_t$. Next, because $\mathbf{x}\_t$ only takes values in a finite domain (line 97) and the absolute value of every $X\_t$ is bounded (equation 21), therefore, $\mathbb{E}[\big| X\_t \big|  | \mathcal{F}\_{t-1}]$ and $\mathbb{E}[\big| Y\_t \big|  | \mathcal{F}\_{t-1}]$ are finite, which suggests that $X\_t$ and $Y\_t=\sum^t\_{s=1}X_s$ are always integrable. This also holds for the case of continuous domains discussed in our response to question (a) above, because we can still apply our algorithm to a finite sub-domain of the continuous domain as discussed there (i.e., $\mathbf{x}\_t$ still only takes values in a finite domain).
> >
> > ---
> >
> > # Theorem 2
> >
> > > (a) It is not convincing to me to have the claim “STO-BNTS-Linear remains asymptotically no-regret as long as the NN is wide enough” based on the result of Theorem 2. That claim means that $\exists N > 0$, s.t., $\forall m > N$ ($m$ is the width of NN), $\lim_{T\rightarrow \infty}\frac{R_T}{T}=0$ when $m$ is fixed. $N$ can be very large but $m$ should be considered a constant. The result of Theorem 2 gives us some intuition on how to choose $m$ when we know the value of $T$ (but even so, $\Omega(T^{24})$ is a very loose bound), but it can not give the asymptotically no-regret conclusion for the finite-width NNs.
> >
> > We agree with you that our claim of “STO-BNTS-Linear remains asymptotically no-regret as long as the NN is wide enough” is not completely accurate.
> > A more accurate statement is that given the value of $T$, as long as the width $m$ is chosen to be large enough such that it satisfies $m=\Omega(T^{24})$, then STO-BNTS-Linear is asymptotically no-regret. We will revise the corresponding claim to make them more accurate.
> >
> >
> > ---
> >
> > > (b,c,d,e) Questions regarding $=$ and $\leq$.
> >
> > In equation (26), you are correct that the first $=$ should be $\leq$. Thank you for pointing this out. We will also correct the other notations and go through all our proofs to make sure that every appearance of $=$ and $\leq$ is accurate. Thanks a lot for carefully checking our proof!
> >
> > ---
> >
> > &#8595; &#8595; &#8595; **Continued below** &#8595; &#8595; &#8595;

---

> > > ### Author Response · Authors · 2022-08-02
> > > **Response to Reviewer an5h (Part 3/3)**
> > >
> > >
> > >
> > >
> > > # Experiments
> > >
> > > > (a) How to set the value of $\sigma$ in the loss function?
> > >
> > > We used $\sigma=0.1$ in all our experiments, which is the default value from the work of [14] (lines 784-787). As shown in our experiments, this value consistently leads to good empirical performances and is hence recommended in practice.
> > >
> > > ---
> > >
> > > > (b) The authors say that they use L-BFGS-B to optimize NN and get the new query, how does this work when the input domain is discrete/categorical or mixed?
> > >
> > > When the input domain is discrete/categorical, instead of using L-BFGS-B, we evaluate the output of the NN for every input in the domain and choose the input with the maximum NN output. When the input domain is mixed, we use standard L-BFGS-B by treating the domain as a continuous one; after finding the input $\mathbf{x}$ that maximizes the output of the NN, we round those discrete inputs to the nearest integer. Both are common practices in the implementation of BO.
> > >
> > > ---
> > >
> > > > (c) The authors say that they also use L-BFGS-B to optimize the acquisition function of Neural Thompson Sampling. What kind of acquisition function is used here?
> > >
> > > We have used the term "acquisition function" for Neural TS to refer to the sampled rewards (which is a function mapping every input to a sampled reward), in order to be consistent with the terminology of BO.
> > > More specifically, for Neural TS, we followed the original algorithm from [35], which samples a reward for every input and then queries the input with the largest sampled reward. When using L-BFGS-B to choose the input to query for Neural TS, every time an input is queried by L-BFGS-B, a reward is sampled at this input and used as the observation for L-BFGS-B.
> > >
> > > ---
> > >
> > > > (d) What kind of implementation do the authors use for GP-TS and GP-UCB? From my previous experiences, I find that even for the same algorithm different implementations will make a large difference in the results. It makes sense to me that standard GP-UCB and GP-TS cannot deal with categorical inputs well, but I am a little bit surprised that they perform so badly in cases with continuous spaces (Figure 3(a)-(c)). After all, according to the recent developments of high-dimensional BO $d\leq 20$ is actually not so high-dimensional.
> > >
> > > For the experiments in Figure 3(a-c), to cater to the higher input dimensions and larger number of BO iterations (500), we used random Fourier features approximation of GPs, which is a common practice to improve the scalability of GPs (especially for GP-TS since it makes it particularly easy to sample a function from the GP posterior). We would like to clarify that GP-UCB and GP-TS underperform significantly only in Figure 3 (a) (likely due to the difficulty of the Lunar-Lander experiment), and they in fact perform well in Figure 3 (b) and (c). Specifically, in Figure 3 (b), GP-UCB (orange curve) performs comparably with our STO-BNTS and STO-BNTS-Linear, and GP-TS also outperforms both Neural TS and Neural UCB; in Figure 3 (c), GP-TS performs the best among all baselines (but not ours), and GP-UCB performs comparably with Neural UCB.
> > >
> > > ---
> > >
> > > > (e) It might not be so fair to compare sequential baselines with STO-BNTS ($B > 1$) under fixed iterations. In that case, STO-BNTS ($B>1$) evaluates the black-box function $B$ times as other methods. It would be more reasonable to compare batched STO-BNTS with other batched baselines such as batched GP-TS (under the same batch size).
> > >
> > > Following your suggestion, we added comparisons with batched GP-TS and batched Neural TS (which simply repeats the query selection of Neural TS independently for $B$ times) using the $12-$D Lunar-Lander experiment (Fig. 3a). The table below shows the results (larger is better), in which all methods use a batch size of $B=4$.
> > >
> > >    | Algorithm | $1$ iterations | $101$ iterations | $201$ iterations | $301$ iterations | $401$ iterations |
> > >    | :--- | :--- | :--- | :--- | :--- | :--- |
> > >    | STO-BNTS  ($B=4$)    | $-124.3 \pm 13.2$ | $267.7 \pm 2.8$ | $278.1 \pm 1.5$ | $280.9 \pm 1.5$ | $282.3 \pm 0.9$ |
> > >    | STO-BNTS-Linear ($B=4$) | $-124.3 \pm 13.2$ |  $257.7 \pm 16.0$ | $264.1 \pm 13.4$ | $265.1 \pm 13.6$ | $267.0 \pm 13.9$ |
> > >    | GP-TS ($B=4$) | $-124.3 \pm 13.2$ | $68.1 \pm 10.4$ | $114.0 \pm 16.0$ | $161.3 \pm 21.3$ | $168.3 \pm 16.2$ |
> > >    | Neural TS ($B=4$) | $-124.3 \pm 13.2$ | $69.7 \pm 7.3$ | $102.0 \pm 22.1$ | $158.9 \pm 21.8$ | $159.6 \pm 21.2$ |
> > >
> > > The results show that with the same batch size, our algorithms still consistently outperform the other baselines with batch evaluations. We will add these results to the paper to better illustrate the advantage of our proposed algorithms.
> > >
> > > ---
> > >
> > > Thank you again for your detailed and valuable feedback. We hope that our additional clarifications and results could help improve your opinion of our paper.

---

> ### Comment · Reviewer_an5h · 2022-08-07
> **Comments on authors response**
>
> Dear authors,
>
> Thank you for your response. I think most responses make sense to me. I just have two follow-up comments.
>
> 1. theorem 2 (a), when $T$ is fixed, I don't think "asymptotic" would be an accurate word to describe the result since now there is no term that will go to infinity.
>
> 2. I am still a little bit confused about using L-BFGS-B on Neural TS. What you do is to use L-BFGS-B to optimize the function $f(x;\theta)$ (2.2 in [35]) first, and if the optimum is obtained, then sample the reward according to line 6 of Algorithm 1 in [35], is that correct? Or like standard TS, just randomly sample $x$ and sample the rewards according to line 6 of Algorithm 1 in [35]?
>
> I have already read all the reviews and comments of this work and I will decide whether it is necessary to adjust my score soon. Besides that, I hope that these comments do help make this work more solid.

---

> > ### Author Response · Authors · 2022-08-09
> > **Response to Further Comments from Reviewer an5h**
> >
> > > 1. theorem 2 (a), when $T$ is fixed, I don't think "asymptotic" would be an accurate word to describe the result since now there is no term that will go to infinity.
> >
> > Thank you for pointing this out. We agree with you that "asymptotic" is inaccurate here, and we will revise the claim of "STO-BNTS-Linear is asymptotically no-regret" (in our original response to your question under Theorem 2. (a)) to "the cumulative regret of STO-BNTS-Linear scales sub-linearly in $T$".
> >
> > ---
> >
> > > 2. I am still a little bit confused about using L-BFGS-B on Neural TS. What you do is to use L-BFGS-B to optimize the function $f(x;\theta)$ (2.2 in [35]) first, and if the optimum is obtained, then sample the reward according to line 6 of Algorithm 1 in [35], is that correct? Or like standard TS, just randomly sample $x$ and sample the rewards according to line 6 of Algorithm 1 in [35]?
> >
> > Sorry for the confusion, we clarify below how Neural TS is implemented in our experiments.
> >
> > **(a)**
> > When the input domain is discrete with a finite number of arms, the original Neural TS goes through every arm $\mathbf{x}$ and samples a reward value for every arm; after that, Neural TS queries the arm $\mathbf{x}$ with the maximum sampled reward value.
> > This can be viewed as **maximizing a function $r(\cdot)$ whose input is an arm $\mathbf{x}$ and output is a sampled reward $r(\mathbf{x})$** (sampled according to line 6 of Algorithm 1 in [35]).
> > We have followed this implementation in our experiments with discrete domains.
> >
> > **(b)** When the input domain is continuous, we modified the above procedure to incorporate L-BFGS-B for optimizing the function $r(\cdot)$. That is, instead of going through every arm $\mathbf{x}$ in the domain (which is infeasible since the domain is continuous), we use L-BFGS-B (from the scipy package) to **maximize the function $r(\cdot)$ whose input is an arm $\mathbf{x}$ and output is a sampled reward $r(\mathbf{x})$** (sampled according to line 6 of Algorithm 1 in [35]).
> > More specifically, in our python implementation, we define a function which takes an $\mathbf{x}$ as the input, samples a reward for this $\mathbf{x}$ according to line 6 of Algorithm 1 in [35], and then returns the sampled reward as the output of the python function; then, we use "scipy.optimize.minimize" (with "method=L-BFGS-B") to minimize the negative of this python function, which returns the input $\mathbf{x}$ to be queried.
> >
> > We will add these further clarifications to the paper.
> >
> > ---
> >
> > We hope that we have clarified these further concerns, and hope that they could help improve your opinion of our work.

---

### Official Review · Reviewer_cYqf · 2022-07-11

**Rating:** 7
**Confidence:** 2
**Soundness:** 3 good
**Presentation:** 2 fair
**Contribution:** 3 good

**Summary:**

The paper proposes a scheme for Thompson sampling based on the neural tangent kernel (NTK). The novelty comes from sidestepping the inversion of the NTK matrix and directly sampling from the equivalent posterior GP using known techniques presented in [14].
Two variants of the algorithms are presented, and regret bounds are presented (and proved in the appendix) for both variants.
An empirical evaluation is included, demonstrating that the proposed methods (in particular Algorithm 1) compare very favorably to a set of baselines on a range of problems.



**Questions:**


Comments/suggestions/comments:

- l111-126 and Algorithm 1: I find the presentation very unintuitive and difficult to follow as specific equations and choices pop out of nowhere. I'd suggest rethinking the structure and order of presentation; a genric short intro to NTK based on [14] might assist in communicating the key steps and justifications which are only partially provided in Appendix A.
-  l111: What does it entail to "....construct an NN f(x;\theta)..." ? I'd suggest adding this step to Algorithm 1 on page 4.
- l 192-202 : I am afraid my intuition/understanding breaks down here.... The claim is that a higher B is better; however, this would imply that with B=T we can achieve the same simple regret in a single iteration? I hope the authors can elaborate and perhaps clarify where the bound is practically useful.
- Figure 2+3. I found reading the figure somewhat tricky due to the split results (figure 2) and reuse of legends for different methods, but I appreciate the difficulty given the number of methods in the comparison. Regardless, I would reconsider how to make the result more accessible.
- Experiments: I would suggest some discussion/justification on the choice and sensitivity of m and L.

Very minor:
- line 317 vs line 314, throughout: I'd suggest being consistent if/when writing out small integer numbers (i.e. 3 vs three).

**Limitations:**

- l391 (final sentence): Personally, I find the final sentence unnecessary, but I appreciate that the authors are trying to meet the formal requirements from NeurIPS.

**Strengths And Weaknesses:**



- The paper addresses a timely problem and is mostly clear except for a few key sections and the use of legends (see question below).
- The basic algorithms are somewhat incremental considering prior work (in particular [14]); however, the theoretical bounds in the TS setting appear to be a substantial contribution (assuming I have not overlooked mistakes in the proofs, etc.). I am slightly unsure about the interpretation/explanations relating to the bounds, but hopefully, the authors can educate me in their rebuttal (see below)
- The empirical results are very encorgining on the included set of relevant problems. Rarely do you see any BO consistenly outperform all other baselines, and I hope the results generalize (I'm keen to try these methods out on my own BO problems...).

---

> ### Author Response · Authors · 2022-08-02
> **Response to Reviewer cYqf**
>
>
> We would like to thank the reviewer for appreciating our contributions and for your insightful feedback.
>
> ---
>
> > l111-126 and Algorithm 1: I find the presentation very unintuitive and difficult to follow as specific equations and choices pop out of nowhere. I'd suggest rethinking the structure and order of presentation; a generic short intro to NTK based on [14] might assist in communicating the key steps and justifications which are only partially provided in Appendix A.
>
> We will revise this section to make it easier to understand, by improving the structure and order of presentation and giving more background on NTK in the context of [14] as you have suggested.
>
> ---
>
> > l111: What does it entail to "....construct an NN f(x;\theta)..." ? I'd suggest adding this step to Algorithm 1 on page 4.
>
> We will follow your suggestion to add this step of constructing an NN to Algorithm 1, and also add the corresponding step to Algorithm 2 accordingly.
>
> ---
>
> > l 192-202 : I am afraid my intuition/understanding breaks down here.... The claim is that a higher B is better; however, this would imply that with B=T we can achieve the same simple regret in a single iteration? I hope the authors can elaborate and perhaps clarify where the bound is practically useful.
>
> Our discussion in this paragraph (lines 192-202) is in fact only applicable when $T$ is large (line 197) compared with the batch size $B$ (i.e., $T\gg B$) and $B$ is a constant independent of $T$. It is no longer applicable if $B$ is large relative to $T$ or if $B$ grows with $T$ (e.g., the case you have mentioned when $B=T$).
>
> The reason is that the regret upper bound is in fact a summation of two terms (lines 185-187): (1) the regrets during initialization which is equal to $2B'T_{\text{init}}$, and (2) the regret upper bound from Theorem 1. As an example, when we choose $C=1$, we have that $T_{\text{init}}=\Theta((B-1)^d)$ (lines 184-185).
> **In the first case** where $T \gg B$ and $B$ is a constant independent of $T$, by ignoring the regrets during initialization which is $\mathcal{O}((B-1)^d)$, the asymptotic regret upper bound simplifies into $R_T=\widetilde{\mathcal{O}}(\sqrt{T}(1+\sqrt{\gamma_T}))$ (lines 187-189). As a result, our discussion in the paragraph in lines 192-202 follows.
> **In the second case** where either $B$ grows with $T$ or $B$ is large relative to $T$, then the regrets during initialization $\mathcal{O}((B-1)^d)$ will no longer be negligible. Therefore, we can no longer simplify the regret upper bound into $R_T=\widetilde{\mathcal{O}}(\sqrt{T}(1+\sqrt{\gamma_T}))$ (lines 187-189). As a result, the subsequent discussion in the next paragraph (lines 192-202), which is based on this simplified regret upper bound, will no longer hold.
> To summarize, our regret bounds are practically useful when $T$ is large compared with $B$ and $B$ is a constant which is independent of $T$.
>
> Thank you for pointing this out. We will revise this section to incorporate our clarification above to avoid potential confusions.
>
> ---
>
> > Figure 2+3. I found reading the figure somewhat tricky due to the split results (figure 2) and reuse of legends for different methods, but I appreciate the difficulty given the number of methods in the comparison. Regardless, I would reconsider how to make the result more accessible.
>
> We will follow your suggestion and try to reorganize the results presented in these figures, to make them more accessible.
>
> ---
>
> > Experiments: I would suggest some discussion/justification on the choice and sensitivity of m and L.
>
> To clarify, we had indeed discussed the choice and sensitivity of the width $m$ and the depth $L$ in different places.
> **Firstly**, regarding the depth $L$, as we discussed in Section 5.2 (lines 297-303), (a) an overly shallow NN usually hurts the performance due to its lack of representation power, and (b) an overly deep NN is also likely to deteriorate the performance due to overfitting. **Secondly**, for the width $m$, as we have discussed in Section 5.1 (lines 275-280) and Section 5.4 (lines 337-338), a larger width $m$ usually improves the performance.
> **Importantly**, we have shown that the choice of $L=2,m=256$ consistently leads to competitive performances in a wide range of experiments (all experiments in Sections 5.2 and 5.3), so we think it can serve as a competitive baseline for many real-world problems.
>
> Sorry for the confusion, and we will follow your suggestion to add a more centralized discussion on the choice and sensitivity of $m$ and $L$ to make them clearer.

---

### Official Review · Reviewer_8ARv · 2022-07-12

**Rating:** 7
**Confidence:** 3
**Soundness:** 4 excellent
**Presentation:** 4 excellent
**Contribution:** 4 excellent

**Summary:**

This paper presents a strategy for performing Bayesian optimization using neural network. The core idea behind the approach is the observation that training a wide neural network (with a slight modification) using the squared loss leads to a sample from the posterior GP with the NTK kernel, thus being equivalent to Thompson sampling. Using this observation, the algorithm turns out to be very simple: train a neural network on the data observed so far and use it as the acquisition function, which is maximized to select the next point to be observed.

The proposed algorithms are experimentally evaluated, and shown to work better compared to other baselines including other neural bandit approaches and GP based approaches.

**Questions:**

How is the performance of the algorithm affected if the unknown function is not sampled from the GP with the NTK kernel, but some other kernel for example a squared exponential kernel?

**Limitations:**

The proposed algorithm seems quite reasonable and can work is a large variety of settings. I can't think of any major limitation of this approach. A minor limitation is that the computational complexity of the algorithm is not completely alleviated. The complexity of each iteration is linear in the number of observations which is better than GPs (cubic). However, the number of parameters of the neural can be quite large which may ultimately lead to a higher overhead in low dimensional problems.

**Strengths And Weaknesses:**

**Strengths**:
- The proposed approach is a much simpler approach compared to past neural bandit approaches, which can be complicated and computationally expensive.
- It comes with theoretical guarantees. It is shown that the cumulative regret grows sub-linearly when the true function is sampled from a GP with the NTK kernel.
- Extensive experimental results show that neural network based approaches perform at par or better than GP based approaches for low-dimensional problems.
- In high-dimensional problems such as with images, neural network based approaches are able to scale where as GP based approaches are not.

**Weaknesses**:
- The current algorithm assumes that the unknown function is sampled from the a GP with the NTK kernel. This can be a strong assumption in many cases.

---

> ### Author Response · Authors · 2022-08-02
> **Response to Reviewer 8ARv**
>
>
> We would like to thank the reviewer for acknowledging our contributions and for your constructive comments.
>
> ---
>
> > The current algorithm assumes that the unknown function is sampled from a GP with the NTK kernel. This can be a strong assumption in many cases.
>
> > How is the performance of the algorithm affected if the unknown function is not sampled from the GP with the NTK kernel, but some other kernel for
> example a squared exponential kernel?
>
> We would like to clarify that in our experiment in Section 5.1, the objective function is indeed sampled from a GP with the squared exponential (SE) kernel (line 251). The results (Fig. 1d) show that our STO-BNTS algorithm (orange curve) can still work well, achieving a comparable performance with GP-TS using the SE kernel (blue curve).
> Moreover, in practice, our algorithm can work in a large variety of settings as you have mentioned, and our experimental results show that our performances are competitive and consistent.
>
> ---
>
> > A minor limitation is that the computational complexity of the algorithm is not completely alleviated. The complexity of each iteration is linear in the number of observations which is better than GPs (cubic). However, the number of parameters of the neural can be quite large which may ultimately lead to a higher overhead in low dimensional problems.
>
> You are correct that if the number of parameters of the NN is excessively large, it can lead to large computational overhead. However, we have shown in our experiments that our methods can achieve competitive performances without the need for overly large NNs (e.g., $L=2,m=256$ consistently work well in all experiments in Sections 5.2 and 5.3). Moreover, BO is usually used to optimize expensive-to-evaluate functions, so the computational cost of training the NN surrogate models can be easily overshadowed by the cost of evaluating the objective function. We will follow your suggestion and add discussions on this to the paper.

---

### Official Review · Reviewer_fxmZ · 2022-07-13

**Rating:** 6
**Confidence:** 4
**Soundness:** 3 good
**Presentation:** 2 fair
**Contribution:** 3 good

**Summary:**

The paper proposes two new algorithms named Sample-Then-Optimize Batch Neural Thompson Sampling (STO-BNTS) and STO-BNTS-Linear by combining the Thompson Sampling strategy and the Neural Network (NN) surrogate model. These methods choose the data to be sampled at each iteration by training an NN surrogate model with randomly initialized parameters and then choose the sampled data as the input data that maximizes the trained NN surrogate model. The proposed methods also support the batch settings. Furthermore, theoretical analysis (in particular the regret analysis) is conducted to understand the convergence property of the proposed methods. Finally, experiments are conducted on multiple real-world experiments including the AutoML and Reinforcement Learning tasks to demonstrate the efficacy of the proposed algorithms.

**Questions:**

•	How are the hyperparameters of the NN surrogate models chosen? Is there any guideline on how to choose these values? Do we need to tune these hyperparameters whenever we train an NN surrogate model?

•	How is the time cost of the proposed algorithms? And how are they compared to existing baselines like GP-TS and Neural-TS?


**Limitations:**

The paper describes some societal impact of the work.

**Strengths And Weaknesses:**

Strengths:

•	The idea of the paper is interesting. I think exploring the research direction of using neural networks as surrogate models in Bayesian optimization is worth to investigate.

•	The theoretical analysis is definitely a strength of this paper. It helps to further understand the behaviours of the proposed algorithms. The theoretical analysis is not just built upon existing analysis in the literature but also introduces new techniques that can be useful for future research in this direction (using neural networks as surrogate models). To the best of my knowledge, the theoretical analysis is sound and reasonable (but note that I could not go deeply into the proof in the appendix, I only judge the theoretical analysis based on my overall knowledge about BO's theoretical analysis)

•	The experiments conducted are quite comprehensive. They include many real-world benchmark problems.

•	The proposed algorithms perform quite well in the benchmark tasks presented in the paper. They are superior to existing baselines.

Weaknesses:

•	The writing could be improved. It took me quite a lot of effort to go back and forth to understand the main idea and the theoretical analysis of the paper.

•	Using neural networks as surrogate models will certainly help to improve the model's accuracy, however, I'm wondering how the hyper-parameters of these NN surrogate models (e.g., the number of layers, the width in each layer, the learning rate) could be efficiently set. From the experimental results, it shows that the performance of BO depends on a lot in the hyperparameters of the NN surrogate models. Finding an optimal set of hyperparameters seem to create another AutoML problem to be solved.

•	I would also like to understand more about the time cost of the proposed algorithms, and how they are compared to the time cost of existing baselines. Training an NN surrogate model for every single input query in the batch probably takes a lot of time.

---

> ### Author Response · Authors · 2022-08-02
> **Response to Reviewer fxmZ**
>
>
> We would like to thank the reviewer for appreciating our contributions and for your insightful comments.
>
> ---
>
> >  How are the hyperparameters of the NN surrogate models chosen? Is there any guideline on how to choose these values? Do we need to tune these hyperparameters whenever we train an NN surrogate model?
>
> > Finding an optimal set of hyperparameters seem to create another AutoML problem to be solved.
>
> For the width $m$ and depth $L$ of the NN surrogate model, we had indeed tested different values in our experiments and derived some guidelines for their selection based on our empirical results. **Firstly**, regarding the depth $L$, as we observed and discussed in Section 5.2 (lines 297-303), (a) an overly shallow NN usually hurts the performance due to its lack of representation power, and (b) an overly deep NN is also likely to deteriorate the performance due to overfitting.
> Therefore, overly shallow and overly deep NNs are discouraged; shallower NNs are preferred for simpler tasks (i.e., simpler objective functions) to prevent overfitting; deeper NNs are encouraged for more complicated tasks to gain enough representation power.
> **Secondly**, for the width $m$, as we have discussed in Section 5.1 (lines 275-280) and Section 5.4 (lines 337-338), a larger width $m$ usually improves the performance. **Importantly**, we have shown that choosing $L=2,m=256$ consistently leads to competitive performances in a wide range of experiments (all experiments in Sections 5.2 and 5.3), so we think it can serve as a competitive baseline for many real-world problems.
>
> For all other hyperparameters of the NN surrogate model except the width and depth, we have used the default values from the work of [14] (lines 784-789). The competitive performances of our methods shown in our experiments suggest that using these default values is a recommended guideline.
>
> In our current paper, we have used fixed values for these hyperparameters as discussed above. However, as you have suggested, using AutoML methods to automatically tune these hyperparameters represents a very interesting extension for our paper, which we indeed plan to explore in future works.
> We will add our discussions here to the paper.
>
> ---
>
> >  How is the time cost of the proposed algorithms? And how are they compared to existing baselines like GP-TS and Neural-TS?
>
> As you have suggested, we additionally compared the wall-clock time (seconds) of different algorithms (with $L=2,m=256$) using the experiment on hyperparameter tuning of RF (Fig. 2a and 2b).
> The table below shows the time/validation error (%) after different numbers of iterations.
>
>    | Algorithm | $5$ iterations | $15$ iterations | $25$ iterations | $35$ iterations | $45$ iterations |
>    | :--- | :--- | :--- | :--- | :--- | :--- |
>    | STO-BNTS      | $84.1/22.0$ | $272.8/20.0$ | $478.1/20.0$ | $700.8/20.0$ | $963.0/20.0$ |
>    | STO-BNTS-Linear | $80.6/22.2$ |  $247.2/22.2$ | $425.6/22.2$ | $613.6/22.2$ | $814.5/22.1$ |
>    | Neural TS | $66.8/21.6$ | $202.5/21.2$ | $338.8/20.7$ | $475.2/20.6$ | $611.6/20.4$ |
>    | GP-TS | $35.3/21.6$ | $106.6/21.1$ | $179.0/21.0$ | $250.7/20.6$ | $322.7/20.6$ |
>
> The results show that our algorithms incur more computational time, however, our STO-BNTS achieves a validation error of $20.0$% after 272.8s, which is better than what Neural TS and GP-TS achieve after 611.6s and 322.7s, respectively.
> Moreover, in practice, BO is usually used to optimize very costly-to-evaluate objective functions. Therefore, our additional computational cost shown above can be easily overshadowed by the time taken to evaluate the objective function.
>
> ---
>
> We will also follow your suggestion to improve the writing of our paper. Thank you again for your constructive comments.

---

> > ### Comment · Reviewer_fxmZ · 2022-08-06
> > **Thank you for the authors' response**
> >
> > Dear authors,
> >
> > Thank you for your response. I appreciate that you have shown the time cost of the proposed algorithms and the baselines. From the table, it seems like the proposed algorithms do not cost much time compared to existing baselines, so I'm happy about that. For the questions regarding the choices of the hyperparameters, I did see those general guidelines the authors mentioned when first reading the paper, however, to me, they're too general to be used as detailed guidelines to set the hyperparameters of the NN surrogate model. And as the authors demonsterated in the paper, the performance of the proposed methods do depend significantly on these hyperparameters, so I think this is still a weakness of the proposed algorithms. For this reason, I decided to keep my score as is, which is 6.

---

> > > ### Author Response · Authors · 2022-08-09
> > > **Thank You for Your Further Comments**
> > >
> > > Thank you for your further comment.
> > >
> > > We agree with you that obtaining a detailed guideline for selecting the hyperparameters of the NN surrogate (specifically, its width and depth) and automatically optimizing them using AutoML methods are interesting extensions to our paper. Thank you for your suggestion, we will indeed explore them in future works.

---

### Meta-Review · Area_Chair_H1qd · 2022-08-29

**Recommendation:** Accept
**Confidence:** Less certain

**Metareview:**

The authors introduced two asymptotically no-regret neural Thompson sampling algorithms. They derived regret upper bounds and showed that they are asymptotically no-regret under certain conditions. They verified their empirical effectiveness with AutoML and reinforcement learning experiments.
All reviewers liked this paper. Please note however, that it is somewhat surprising that in some cases that standard GP-UCB and GP-TS competitors performed very badly. One of the reviewers reproduced the Lunar-Lander experiments in BoTorch and achieved much better performance for these competitor methods.


**Award:**

No

---

### Decision · Program_Chairs · 2022-09-14

Accept